# Agricultural non-point source pollution and health of the elderly in rural China

**Ying Wang**[1], **Hang Xiong**[1,2], **Chao Chen**[1,2]*

**1** College of Economics and Management, Nanjing Agricultural University, Nanjing, Jiangsu, P.R.China,
**2** China Center for Food Security Studies, Nanjing Agricultural University, Nanjing, Jiangsu, P.R.China

* cchen@njau.edu.cn

**Data Availability Statement:** The data underlying the results presented in the study are available from https://opendata.pku.edu.cn/dataverse/CHADS;jsessionid=6a16c8c5d8c20925c51b93da42fc.

## Abstract

Large input and high loss of chemical fertilizer are the major causes of agricultural non-point source pollution in China. Employing fertilizer loss and micro-health data, this paper analyzes the effects of chemical fertilizer loss on the health of rural elderly and the medical cost in China. Results of the difference-in-differences (DID) method indicate that one kg/ha increase in fertilizer loss alters a key medical disability index (Activities of Daily Living) by 0.0147 (0.2 percent changes) and the number of diseases by 0.0057 for rural residents of 65 and older. This is equivalent to CNY 316 million (USD 45 million) at national medical cost. Furthermore, the age of onset is younger in regions with higher fertilizer loss. One kg/ha increase of fertilizer loss advances the age of onset by 0.267 year, which will cause long-term effect on public health. Our results are robust to a variety of robustness checks.

## 1. Introduction

China experienced rapid growth in chemical fertilizer use, particularly after 2004, when the Chinese government shifted from taxing agriculture to subsidizing agricultural programs. The unit area chemical fertilizer usage is nowadays 4 times more than the world average [1]. Agricultural non-point source pollution is a growing concern in China because these pollutions had become the major cause of water pollution [2]. Agricultural non-point source pollution has also been found to be an important factor affecting health [3–6]. Specifically, the elderly might have a higher health risk comparing with the young people when facing environmental exposure. A higher susceptibility and higher mortality rates could be observed among the adults over the age of 75 years compared with the younger ages [7–9]. The differences in health risk might be because of physiological changes associated with aging [10]; the age-related behavior trends varied between younger and older adults, which might affect older adults' exposures [11]; other factors include socio-economic and nutritional status [12]. China had the most serious aging problems because 70% of elderly lives in rural areas [13]. The elderly residents might most be affected by agricultural non-point pollution. Another reason we focus on the rural elderly is that the health status, medical facility and diseases awareness of rural elderly are not as good as those in cities [14, 15]. This paper complements the existing literature by proving recent evidence on the effects of agricultural non-agricultural pollution. Air

**Funding:** This article was sponsored by the Priority Academic Program Development of Jiangsu Higher Education Institutions (PAPD). The funders had no role in study design, data collection and analysis, decision to publish, or preparation of the manuscript.

**Competing interests:** The authors have declared that no competing interests exist.

pollution as a leading problem for public health was broadly studied in agricultural non-point pollutions [16, 17]. The work most closely related to this paper was Brained and Menon (2014) [4], which assessed the impact of fertilizer runoff on infant and child in India. We complement their study by emphasizing another vulnerable group, i.e., older adults, who tend to stay in the countryside during urbanization and are directly affected by the loss of chemical fertilizer. Another work by Lai (2017) [5] assessed the health effect of pesticide by using the pesticide cost. In order to eliminate the influence of price, we use the amount of chemical input and loss, which are more accurate. Though both chemical fertilizers and pesticides can cause water pollution, they work in different ways, fertilizer are directly spread in the soil, which is more likely to affect by surface and groundwater runoff. This paper emphasizes on non-occupational fertilizer exposure because the farmers who are directly exposure to chemical fertilizer are only a small proportion of the population and had been studied [18–20]. Many more population may be at risk because of agricultural externality.

The inefficient use of chemical fertilizer has caused serious non-point source pollution. Fertilizers leaded to "groundwater contamination, eutrophication of fresh water and estuarine ecosystems, tropospheric pollution related to emissions of nitrogen oxides and ammonia gas, and accumulation of nitrous oxide" [21]. A 10% increase in using nitrogen fertilizer and phosphorus fertilizer leaded to a 1.525% and 1.374% increase of the concentration of nitrogen and phosphorus in water, respectively [22]. In China, agricultural non-point source pollution has exceeded industrial pollution in water. The chemical oxygen demand (COD) emission of agricultural pollution sources was 1.8 times of the industrial sources [23]. After 2004, in particular, China abolished agricultural tax and issued a series of agricultural support policies to promote agricultural production, which encouraged the excessive application of agricultural fertilizer. From 2004 to 2015, the amount of chemical fertilizer applications was increased by 30% (i.e., 1.386 million tons) [24], while the increase in agricultural produce output was far less. As a result of the formation of nitrite compounds and the change in soil environmental conditions, chemical fertilizer loss might lead to chronic diseases, nervous system diseases, and organ diseases [25–27], and eventually affect human health and medical cost (Fig 1). Amines and amides in chemical fertilizers and sewage could cause digestive tract diseases, esophageal

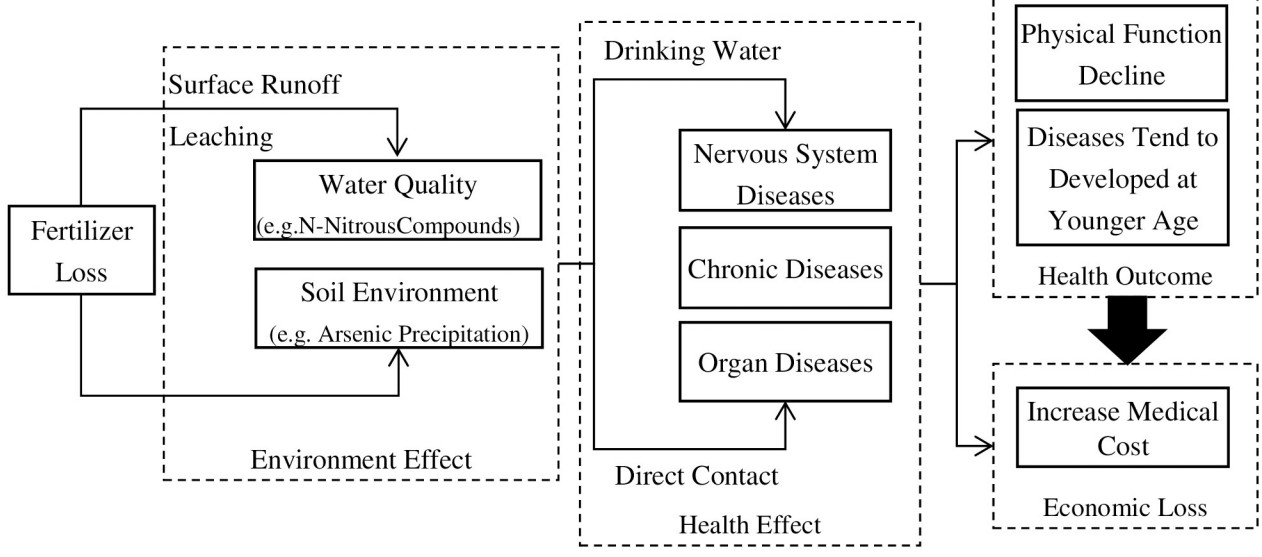

**Fig 1. Effects of chemical fertilizer loss on health.**

cancer, liver cancer, stomach cancer, esophagus cancer, and other cancers [28–30]. The fertilizer, which were soluble in water and had stable properties, were transported by rivers over long distances and large ranges, and enter the human body via drinking water and other ways to synthesize nitrosamines, especially nitrosamines. Drinking water or eating vegetables with a high nitrate content may lead to thyroid cancer, hypertension, neural tube defects (NTDs) [31, 32], and various chronic diseases, such as skin diseases, lung cancer, bladder cancer, and cardiovascular disease [33]. Based on the background of China's agricultural support policy, this paper verifies the effect of chemical fertilizer loss from the soil on the health of the rural elderly, and comprehensively analyzes the health damage from the perspectives of daily living ability, disease risk, and disease age onset. The economic cost of this health damage is further calculated. More specifically, we propose 3 hypotheses: (1) Fertilizer loss can cause physical damage, increase the health risk and alter the daily living ability of the elderly. (2) Because of the physical function, fertilizer loss may lead the individuals to be ill younger. (3) Because of the health damages, the medical cost will be higher in high-loss regions comparing the regions with lower fertilizer loss.

The rest of the paper is organized as follows. Section 2 provides data description and summary statistics. Section 3 presents empirical models and results and Section 4 concludes.

## 2. Data and summary statistics

### 2.1 Data collection and variables

The analysis is accomplished using the difference-in-differences method with the national dataset Chinese Longitudinal Healthy Longevity Survey (CLHLS) and publicly published yearbooks.

The individual health data come from the Chinese Longitudinal Healthy Longevity Survey (CLHLS). We choose the CLHLS database because of its richness in health and demographic information, its nationwide sampling areas, and its time span which covers the year of the agricultural policy change. The CLHLS was conducted in a randomly selected half of the total counties and cities in 22 provinces, covering 85% of the total population in China. Its eight waves (1998, 2000, 2002, 2005, 2008, 2011, 2014 and 2018) surveyed the cohort of 65 years and older. In order to exclude possible health links to waterborne bacteria, we only use the data from 2000 to 2014 because the survey question asking about boiling water only began in 2000. China implemented the "fertilizer reduction" policy in 2015. In order to eliminate this policy effect, we did not use the data after 2015. The survey combines an in-home interview and a basic physical examination. Extensive information was collected on demographic characteristics, family and household characteristics, lifestyles, diet, psychological characteristics, health, disability, socioeconomic conditions, etc. Fertilizer input data come from the China Rural Statistical Yearbook (1999–2015) and other regional variables come from the China Statistical Yearbook (1999–2015). Chemical fertilizer loss index come from data collection of the first national survey of pollution sources (2011).

**2.1.1 Health outcome variables.** Several health indicators are used to determine the health status of the elderly, including the index of Activities of Daily Living (ADL), the numbers of diseases related to fertilizer loss, and age of onset, based on existing medical and toxicology literature. The ADL index measures the dependence of the elderly and physically inactive, and is also known as the "disability" index [34]. The ADL index is constructed by asking participants whether they need help with six basic activities (namely bathing, dressing, eating, going to the toilet, indoor movement, and controlling defecation). Each item has three choices: "can do without help" (corresponding score is 1), "need help" (corresponding score is 2), and "need full help" (corresponding score is 3). Considering all six items, the ADL index

 

has a final score ranging from 6 to 18. Higher scores indicate poorer health conditions among the elderly.

The number of diseases measures how many types of diseases are associated with fertilizer loss that a sample individual suffered form. Although the CLHLS database investigated 16 diseases in the elderly, including hypertension, heart disease, cancer, stroke, cardiovascular and cerebrovascular diseases, bronchitis, emphysema, asthma, and pneumonia, only three diseases —hypertension, cancer, and gastrointestinal ulcer—were related to fertilizer loss in the existing medical literature [27, 28, 33].

The age of onset refers to the age at which a sample individual first suffered from any of the three diseases (i.e., hypertension, gastrointestinal ulcer, or cancer).

**2.1.2 Individual and household characteristic variables.** Individual and household characteristic variables include information on socio-economic characteristics, health behavior and dietary pattern. Socio-economic characteristics include age, gender, people living together, income status, basic health condition and hospitalization. Health behaviors measure whether the respondents smoke, drink or do any exercise (with options of "Yes" or "No"). Dietary patterns measure the frequency with which respondents eat meat, fish, egg, and salt-preserved vegetables. Options for each item are "rarely or never", "not every month, but occasionally", "not every week, but at least once per month", "not every day, but at least once per week", "almost every day", with scores from 1 to 5, respectively. The questionnaire also asks the sources of respondents' drinking water. Options include drinking water sources from a well, a river of lake, a spring, a pond or pool, and tap water. A binary variable (Water) is constructed, which equals one if respondents drink surface water, i.e. water from a river or lake or a pond of pool, and equals zero otherwise.

**2.1.3 Provincial level variables.** Provincial level variables include fertilizer loss, fertilizer use, organic fertilizer use, pesticide use, industrial pollution index, the numbers of hospital and provincial Gross Domestic Product (GDP). The calculation method for fertilizer loss intensity is based on literature [35]. Base on the unit investigation method and the inventory analysis method, the fertilizer loss coefficient is used to calculate the total nitrogen and phosphorus emissions; then the total emission and emission intensity of fertilizer non-point source pollution are calculated. The calculation formula is:

$$E = \sum E_{ij} = \sum C_{ij} \times \eta_{ij} \tag{1}$$

$$I = E/A \tag{2}$$

where $E$ measures the total emission of fertilizer loss; $E_{ij}$ measures the amount of the pollutant $j$ produced and the loss into water by province $i$; $C_i$ measures the index statistics of pollutant $i$ (compound fertilizer is converted into total nitrogen 40% and phosphorus pentoxide 32%); $\eta_{ij}$ measures the loss coefficient; $A$ represents the crop areas; and $I$ represents pollution emission per unit area, i.e., pollution intensity (unit: kg/ha).

The industrial pollution index ($Indpollution_i$) controls industrial activities, which is defined in Eq (3) according to existing literature [36, 37]. GDP measures general economy and wealth conditions. The number of hospitals controls the level of medical service. Because the health data are observed once three years, all provincial variables are assigned the average of the last three years.

$$Indpollution_i = \frac{pv_{i1} + pv_{i2} + pv_{i3}}{3}, pv_{ij} = p_{ij} \bigg/ \sum_{i=1}^{n} \frac{p_{ij}}{n} \tag{3}$$

where $p_{ij}$ measures the pollutant $j$ of province $i$, $n$ is the total number of provinces, $pv_{ij}$

measures the emission index of pollutant $j$ in the province $i$ relative to the national average level. The larger value means the higher pollution level. $pv_{i1}$, $pv_{i2}$, $pv_{i3}$ represents industrial wastewater, industrial sulfur dioxide and industrial dust, respectively.

## 2.2 Statistical analyses

**2.2.1 Sample summary.** The statistical descriptions of all variables used in the regression are summarized in Table 1.

**2.2.2 Fertilizer loss and health outcome after agricultural support policies.** China shifted from taxing agriculture to subsidizing agricultural programs since 2004 to maintain food security and self-reliance. In 2004, authorities introduced three subsides targeted at grain producers: a direct payment for grain producers, a subsidy for improved seed varieties and a partial rebate for farm machinery purchases. Several other support programs were introduced such as a general-input subsidy, price floors for wheat and rice, reform of the grain marketing system and transfer payment to grain counties since 2004, which has been accompanied with

**Table 1. Variable description and summary statistics.**

| Variables | Description | Obs. | Mean | Std.Dev. | Min | Max |
|---|---|---|---|---|---|---|
| **Household Variables** | | | | | | |
| Age | Age in years | 39093 | 85.83 | 10.83 | 65 | 109 |
| Male | Male = 1; Female = 0 | 39093 | 0.44 | 0.50 | 0 | 1 |
| Co-residence | Nursing home = 1; Alone or Spouse = 2; Child = 3; Others = 4 | 39093 | 2.62 | 0.57 | 1 | 4 |
| Income_cost | If income support daily cost (Yes = 1; No = 0) | 39093 | 0.78 | 0.41 | 0 | 1 |
| Illness | Number of 15 chronic diseases | 39093 | 0.85 | 1.06 | 0 | 1 |
| Hospitalization | The times of hospitalization in two years | 39093 | 0.26 | 0.79 | 0 | 30 |
| **Health Behaviors** | | | | | | |
| Smoke | Currently smoke (Yes = 1; No = 0) | 39093 | 0.33 | 0.47 | 0 | 1 |
| Drink | Currently drink (Yes = 1; No = 0) | 39093 | 0.31 | 0.46 | 0 | 1 |
| Exercise | Currently exercise (Yes = 1; No = 0) | 39093 | 0.27 | 0.44 | 0 | 1 |
| Dietary Pattern | | | | | | |
| Meat | The frequency of meat consumption | 39093 | 3.95 | 1.03 | 1 | 5 |
| Fish | The frequency of fish consumption | 39093 | 3.36 | 1.15 | 1 | 5 |
| Egg | The frequency of egg consumption | 39093 | 3.92 | 1.07 | 1 | 5 |
| Salt_vege | The frequency of salt-preserved vegetable consumption | 39093 | 3.27 | 1.38 | 1 | 5 |
| Boiled water | If drink boiled water (boiled water = 0; not boiled water = 1) | 39093 | 0.057 | 0.23 | 0 | 1 |
| Water | Drinrking water source (Surface water = 1; Tap water = 0) | 39093 | 0.14 | 0.35 | 0 | 1 |
| **Provincial Variables** | | | | | | |
| Floss | Fertilizer loss intensity (kg/ha/year) | 39093 | 7.34 | 3.76 | 0.60 | 14.82 |
| Finput | Fertilizer input intensity (kg/ha/year) | 39093 | 354.80 | 92.47 | 128.41 | 704.32 |
| Pinput | Pesticide input intensity (kg/ha/year) | 39093 | 12.62 | 6.56 | 2.28 | 22.26 |
| OFinput | Organic fertilizer input intensity (kg/ha/year) | 39093 | 286.53 | 361.18 | 11.56 | 1997.00 |
| Hospnum | The number of hospital per province (million) | 39093 | 0.02 | 0.02 | 0.01 | 0.08 |
| LnGDP | Logarithm of Provincial Gross Domestic Product | 39093 | 9.10 | 0.91 | 7.32 | 11.04 |
| Indpolltion | Index of industrial pollution (lower = better) | 39093 | 1.46 | 0.55 | 0.45 | 2.95 |
| **Health Variables** | | | | | | |
| ADL | Activities of Darly Living score (lower = better) | 39093 | 6.66 | 1.80 | 6 | 18 |
| No. of diseases | The number of illness caused by fertilizer loss (lower = better) | 39093 | 0.25 | 0.48 | 0 | 3 |
| Age | Age of suffering from illness caused by fertilzer loss (lower = worse) | 9503 | 83.70 | 10.06 | 65 | 109 |
| Mescost | Medical cost(RMB) | 39093 | 2768.29 | 11676.91 | 0 | 199996 |

**Table 2. Comparison of high-loss-areas and low-loss areas by agricultural support policies.**

| Variables | Before agricultural Support Policies | | | After Agricultural Support Policies | | | 2nd difference (7) = (6)-(3) |
|---|---|---|---|---|---|---|---|
| | Low loss (1) | High loss (2) | 1st difference (3) = (2)-(1) | Low loss (4) | High loss (5) | 1st difference (6) = (5)-(4) | |
| ADL index | 6.823 | 7.022 | 0.199 | 6.822 | 7.108 | 0.287 | 0.088 |
| No. of diseases | 0.160 | 0.211 | 0.051 | 0.255 | 0.340 | 0.085 | 0.035 |
| Input intensity (kg/ha) | 239.562 | 358.413 | 118.851 | 317.519 | 457.163 | 139.644 | 20.793 |
| Total loss (10000 tons) | 1.689 | 6.717 | 5.028 | 2.023 | 7.350 | 5.327 | 0.299 |
| Loss intensity (kg/ha) | 2.905 | 8.599 | 5.694 | 3.491 | 9.226 | 5.735 | 0.041 |

Notes:

The fertilizer loss is a continuous variable, and we dichotomized fertilizer loss by its pre-policy mean to compare the health effect in different regions. The average value of fertilizer loss is the national average of fertilizer loss; it is equal to 5.36kg/ha.

We showed the fertilizer input and fertilizer loss in S1 Fig.

greater fertilizer use and fertilizer loss. Although these programs had bolstered farmer' production incentives and crop production had increased continuously since 2004 [38, 39], the programs also might l distort of agricultural factors and have negative effects on soil and water quality [40]. Table 2 reports the differences of health outcomes and fertilizer loss between regions with high versus low intensity of fertilizer loss. We report the differences of ADL index, the numbers of induces diseases, intensity of fertilizer input, amount of fertilizer loss and intensity of fertilizer loss before implementing agricultural support policies in column (3). The differences after the agricultural support policies are in column (6). We can observe the widening gap in health outcomes and fertilizer loss and input in column (7) between regions with high and low fertilizer loss after the agricultural support policies. Based on preliminary data analysis, the agricultural subsidy policies resulted in greater loss of chemical fertilizer in high-loss areas.

**2.2.3 Age of onset and fertilizer loss.** Among the CLHLS, there are three diseases (namely hypertension, gastrointestinal ulcers and cancer) caused by fertilizer loss. Age of onset refers to the age when one sample suffers from any of the three diseases for the first time. Fig 2

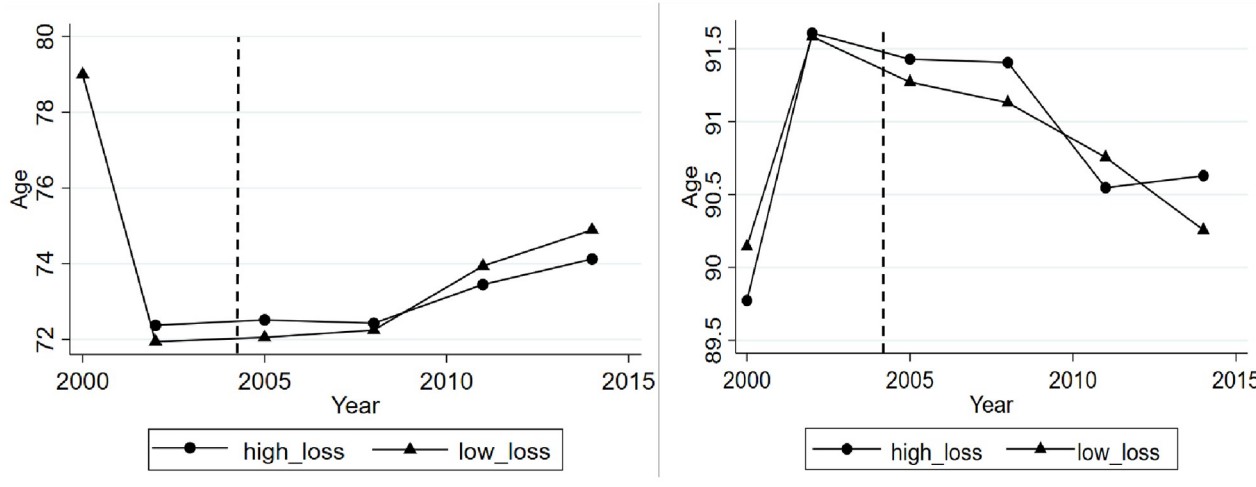

**Fig 2. Chemical fertilizer loss and age of onset.**

compares age of onset across age groups. A significant difference of age of onset can be observed between the regions with high versus low intensity of fertilizer loss in younger-old groups (< 80 years) in Panel A of Fig 2. The age of onset is not significantly different in the oldest-old samples.

## 3. Empirical models and results

This section investigates the health effect of fertilizer loss on the local elderly population. Preliminary data analysis reports the health outcomes are different in regions with high verse low intensity of fertilizer loss after agricultural subside policies in 2004. Further, we follow a difference-in-differences (DID) framework to compare health outcomes between people in regions with different intensities of fertilizer loss before and after 2004, when China shifted from taxing agricultural outputs to subsidizing agriculture. The medical cost is estimated as well. To facilitate comparisons across age groups, we explore heterogeneous effects by splitting the sample in various age groups and reporting estimates of $\alpha_1$ and $\beta_1$ for these sub-samples. Then we conduct several robustness checks to assess the validity of our results. The difference-in-differences model is specified as:

$$Health_{it} = \alpha_0 + \alpha_1 Fertilizerloss_{it} \times Time_{it} + \alpha_2 Fertilizerloss_{it} + \alpha_3 Time_{it} + \alpha_4 X_{it} + \gamma_t + \mu_i + \varepsilon_{it} \quad (4)$$

$$Health_{it} = \beta_0 + \beta_1 Area_{it} \times Time_{it} + \beta_2 Area_{it} + \beta_3 Time_{it} + \beta_4 X_{it} + \gamma_t + \mu_i + \varepsilon_{it} \quad (5)$$

$$Age_{it} = \delta_0 + \delta_1 Fertilizerloss_{it} \times Time_{it} + \delta_2 Fertilizerloss_{it} + \delta_3 Time_{it} + \delta_4 X_{it} + \gamma_t + \mu_i + \varepsilon_{it} \quad (6)$$

$$Age_{it} = \theta_0 + \theta_1 Area_{it} \times Time_{it} + \theta_2 Area_{it} + \theta_3 Time_{it} + \theta_4 X_{it} + \gamma_t + \mu_i + \varepsilon_{it} \quad (7)$$

where $Health_{it}$ denotes the health outcome; $Age_{it}$ denotes the age of onset; $\alpha_1$, $\beta_1$, $\delta_1$ and $\theta_1$ identify the effects of fertilizer loss; $Fertilizerloss_{it}$ denotes the fertilizer loss of the region where individual $i$ resides; $Area_{it}$ is a dummy variable equaling 1 if the individual is in high loss areas, and 0 otherwise; $Time_{it}$ equals 1 starting in 2004 and 0 otherwise; $X_{it}$ is a vector of control variables including individual, household, and provincial characteristics such as age, gender, health behavior, dietary habits, industrial pollution and hospitals, etc.; $\mu_i$ is a fixed effect unique to individuals $i$; $\gamma_t$ is a time effect common to all individuals in year $t$; and $\varepsilon_{it}$ is an error term.

The relationship between medical cost and health status can be identified in the following Equation:

$$Y_{it} = \alpha_0 + \alpha_1 ADL_{it} + \alpha_2 X_{it} + \gamma_t + \mu_i + \varepsilon_{it} \quad (8)$$

where $Y_{it}$ represents medical costs, and other variables are defined as before.

### 3.1 Basic regression results of health

**3.1.1 Effect of fertilizer loss on health.** Table 3 reports the $\alpha_1$ and $\beta_1$ coefficients in Eqs (4) and (5) for outcomes related to ADL index and No. of diseases. Estimated coefficients in column (1) of Table 3 indicate that one kg/ha increase of chemical fertilizer loss is associated with increased ADL daily life index of 0.0147 (0.2 percent). From column (4) one kg/ha increase of fertilizer loss is associated with 0.0057 raise in the number of diseases (0.5 percent). These results indicate that the fertilizer loss increase the elderly's health risks and support diseases significantly. For different regions, estimates for $\beta_1$ in columns (2) and column (5) of Table 3 indicate a reduction in the ADL index and the number of diseases for high-loss area verse low-loss areas. Compared with low-loss areas, the ADL index of the rural elderly is

Table 3. Basic estimates ADL index and diseases.

| | ADL | | | No. of diseases | | |
| | DID | | TWFE | DID | | TWFE |
| | (1) | (2) | (3) | (4) | (5) | (6) |
|---|---|---|---|---|---|---|
| Fertilizerloss*Time | 0.0147** | - - | - - | 0.0057*** | - - | - - |
| | (0.0060) | - - | - - | (0.0014) | - - | - - |
| Higharea*Time | - - | 0.1222*** | - - | - - | 0.0487*** | - - |
| | - - | (0.0461) | - - | - - | (0.0114) | - - |
| Fertilizer loss | - - | - - | 0.1088** | - - | - - | 0.0211* |
| | - - | - - | (0.0432) | - - | - - | (0.0126) |
| Control | YES | YES | YES | YES | YES | YES |
| Year FE | YES | YES | YES | YES | YES | YES |
| ID FE | YES | YES | YES | YES | YES | YES |
| Observations | 32165 | 32165 | 38908 | 32329 | 32329 | 38680 |
| R-squared | 0.5570 | 0.5582 | 0.3183 | 0.6906 | 0.6907 | 0.3243 |

Notes:

*, **, *** are significant at the level of 10%, 5% and 1% respectively.

The estimated results without control variables were in S1 Table in S1 File.

Considering the nonlinear health factors, we further estimated the health effects of fertilizer loss using propensity score matching (PSM) difference-in-differences (DID) model. The results were in S2 Table in S1 File.

0.1222 higher in areas with high chemical fertilizer loss, while the number of diseases is 0.0487 higher. We also use the two-way fixed effects model to estimate the health effect of fertilizer loss. Column (3) and Column (6) of Table 3 report the coefficients for outcomes related to ADL index and No. of diseases. Estimated coefficients show that one kg/ha of fertilizer loss increase is associated with increased ADL daily life index of 0.1088. Column (6) shows that one kg/ha of fertilizer loss increase is associated with 0.0211 raise in the number of diseases. The health effect of agricultural fertilizer input not only accrues in farmers who are likely to receive a fertilizer loss via direct exposure. Many more proportions of the population are at risk of fertilizer loss because of exposure to contaminated water and soil. Agricultural support policies play an important role in the fertilizer loss, and the agricultural support policies may cause heterogeneous fertilizer loss between regions because of the different subsides levels. These may cause the biased result of TWFE. DID model can avoid this problem, and we control the time effect and individual effect in our regressions.

**3.1.2 Heterogeneous health effects.** In S3 and S4 Tables in S1 File, we explore the heterogeneous effects by splitting the samples in different age groups and reporting estimates of $\alpha_1$ for these sub-samples. First, the oldest olds ($\geq$80years) were discussed in previous studies [41], because the oldest olds had greater risk of falling and hospitalization [42–44]. We divide the samples at age into the younger age samples (<80years) and the oldest olds samples ($\geq$80 years). In Columns (1) and columns (2) of S3 Table in S1 File, a positive and significant estimate of $\alpha_1$ can be observed in the oldest olds. The estimates of $\alpha_1$ are positive and significant for both age groups (columns (3) and columns (4) of S3 Table in S1 File). The fertilizer loss will increase the risk of diseases in both the younger olds and the oldest olds. The grouped estimates for effect of fertilizer loss on ADL and diseases of the oldest old are significantly distinguishable from the younger olds at the 1% level. We also divide the samples into younger olds (65years-75years), middle olds (75-85 years) and the oldest olds ($\geq$85years) [45–47]. The differences between younger and middle olds are not significant in both ADL and No. of diseases.

 

**Table 4. Estimation of fertilizer loss on age of onset.**

| | Age of Onset | |
| --- | --- | --- |
| | (1) | (2) |
| Fertilizerloss*Time | -0.2670* | |
| | (0.0161) | |
| Higharea*Time | | - 0.3753** |
| | | (0.1258) |
| Control Variables | YES | YES |
| Year FE | YES | YES |
| ID FE | YES | YES |
| Observations | 9503 | 9503 |
| R-squared | 0.9822 | 0.9822 |

Notes:

*, **, *** are significant at the level of 10%, 5% and 1% respectively.

For oldest old, the coefficient f is larger than for younger and middle olds, and the difference is significant.

**3.1.3 Effect of fertilizer loss on age of onset.** This study further explores whether fertilizer loss will lead to the elderly be sick in younger age. Though from Fig 2, the age of onset of elderly in areas with high-loss of chemical fertilizer are younger, in order to understand the relationship between fertilizer loss and the age of onset, we use the difference-in-differences model for empirical analysis. Estimated coefficient in column (1) of Table 4 indicates that one kg/ha increase of fertilizer loss advances the age of onset by 0.2670 year. Compared with low-loss areas, the age of onset declined by 0.3753 in high-loss areas.

## 3.2 Results of health conditions on medical cost

In this section, we continue to explore these results by translating estimated effects into expected monetary losses. We do this by exploiting the relation between ADL and medical cost in Eq (8). Estimated coefficient in column (1) of Table 5 indicates that each additional unit increase in ADL index increases the medical cost by CNY 244.6 (USD 34.94). Columns (2) and (3) report the $\lambda_1$ coefficient in Eq (8) in high-loss and low-loss areas, respectively. The

**Table 5. Effect of health on medical cost.**

| | Medical Cost | | |
| --- | --- | --- | --- |
| | Overall samples | High-loss area | Low-loss area |
| ADL index | 244.60*** | 331.94*** | 137.30** |
| | (55.81) | (93.54) | (63.58) |
| Control Variables | YES | YES | YES |
| Year FE | YES | YES | YES |
| ID FE | YES | YES | YES |
| Observations | 23087 | 19875 | 15167 |
| R-squared | 0.0709 | 0.0226 | 0.0121 |
| p | | | 0.0853* |

Notes:

*, **, *** are significant at the level of 10%, 5% and 1% respectively.

estimates show that there is a significant difference in medical cost between high loss areas and low loss areas. Every one-unit ADL increase is associated with a higher medical cost of CNY 194.64 (USD 27.81) in high loss areas. The health cost per unit loss of chemical fertilizer (1kg/ha) is about CNY 3.64 (USD 0.51). Though agro-chemical use is associated with significantly greater agricultural output value, but many studies showed that agro-chemical also damaged human health. Indirectly, medical cost increases and labor supply loss could be observed because of illness [48]. Similar to our study, Lai suggested that a 10%(CNY 0.2/USD 0.03) increase in rice pesticide use will add 168.8 and 55.89 million dollars to medical costs and off-spring's human capital losses, respectively [6]. According to the sixth national census, there were about 87.91 million rural elderly people (over 65 years of age) in China. We estimate the economic loss of fertilizer loss, an increase of one kg/ha of fertilizer loss and fertilizer input will add CNY 316 (USD 45) million and CNY 21.5 (USD 3.1) million to national medical cost, respectively.

### 3.3 Robustness check

**3.3.1 Parallel trend test.** In order to test whether the time staggered entry events of agricultural policy are effective exogenous shocks, we conduct a parallel test based on Eq (9):

$$H_{it} = \alpha + \sum_{k \geq -2, k \neq -1}^{k=4} \delta_k D_{it,k} \times Fertilizerloss_{it,k} + \lambda X_{it} + \gamma_t + \mu_i + \varepsilon_{it} \tag{9}$$

where $D_{it,k}$ is a binary variable that takes value of 1 when agricultural support policies are implemented. We use $k = -1$ to denote the base year. S2 Fig, Panel A and Panel B plots estimates of ADL index and No. of diseased, respectively. The pre-period coefficients are statistically indistinguishable from zero and there is a sharp, statistically significant and sustained change after the policy implementation. The post-coefficients are significant. We also adopt a binary variable that takes value of 1 when the individuals in high loss areas. We showed the estimates coefficients in S2 Fig Panel C and Panel D.

**3.3.2 Placebo test.** In order to further test whether the results are driven by unobservable factors at the individual or provincial level, referring to the methods of existing literature [49, 50], this study conducted a placebo test by randomly assigning high and low chemical fertilizer loss areas. Specifically, 9 out of 19 provinces are randomly selected as the experimental group, assuming that these 9 provinces are areas with high loss of chemical fertilizer, and other regions are the control group (areas with low loss of chemical fertilizer). In this study, random sampling is carried out 1000 times, and the benchmark regression is performed according to Eq (5). S3 Fig reports the average value of the ADL index and the number of induced diseases after 1000 random distributions. The mean value of the 1000 random sample regressions is almost 0. The placebo test shows that our basic estimation results are unlikely to be driven by unobservable factors at the individual or provincial level.

**3.3.3 Effect of fertilizer input on health.** In order to verify the robustness of the research conclusions, the application intensity of chemical fertilizer and the areas with high and low chemical fertilizer application intensities are used to test the robustness. According to the criteria of ecological counties in China (the application intensity of chemical fertilizer should not exceed 250 kg/ha), the areas with a chemical fertilizer application intensity exceeding 250 kg/ha are defined as high input areas, and the areas with a chemical fertilizer application intensity not exceeding 250 kg/ha are defined as low input areas. By using the chemical fertilizer application intensity, we re-estimated Eqs (4) and (5). S5 Table in S1 File shows the results of the model. Based on the model results, the research results are robust when compared to the

                                                   

benchmark regression results. For every one kg/ha increase in fertilizer input intensity, the ADL index will increase by 0.001 and the number of induced diseases will increase by 0.0001.

**3.3.4 Effect in different areas.**   In order to verify the robustness of the analysis of this study, different samples were used. This paper focuses on the health effect of agricultural fertilizer loss of the rural elderly. Therefore, samples from major agricultural-production areas, major rice-producing areas and water resource-rich areas are used to carry out the basic regression of Eqs (4) and (5). The results (S6 and S7 Tables in S1 File) show that fertilizer loss leads to a significant increase both on ADL index and the number of diseases in major agricultural-producing areas, major rice-producing areas and water resource-rich areas, which are basically the same comparing with results in Table 3.

## 4. Conclusions and discussion

China's agricultural fertilizer input is facing a problem of "high input, high loss, and high pollution", which has been aggravated by agricultural support policies since 2004. The problem of the "Three Highs" of chemical fertilizer use has caused pollution to the environment and health damages to rural residents. In this paper, the chemical fertilizer loss data and CLHLS micro-health data are used to verify the health effect of fertilizer loss on the rural elderly in China. Nearly 60% elders live in rural areas in China and the number of the elders might increase to 450 million (nearly one third of the total population) by 2050 [51]. Fertilizer loss may become a greater challenge to pension and health care in rural China.

To improve the health status of elderly in rural areas, the government can act by using agricultural policy and public health policy tools. (1) fertilizer loss can be reduced by innovating agricultural production technology research and promoting technology adoptions, e.g. soil testing, formula fertilization technology, slow-release fertilizer technology, deep tillage machinery promotion, and straw returning technology. (2) the government should improve soil quality to increase nutrient preserving capability of soil, such as soil with soil improvement technology. (3) the government needs to focus on the rural drinking water, and to improve the rural elderly's awareness of drinking safety and the risk of health damages because of chemical fertilizer loss.

We explored the effect of agricultural non-point source pollution on public health with fertilizer loss as a representative. Although animal husbandry pollution was regarded as point source pollution in our study, free range livestock farming in rural China might also cause non-point pollution. Because of data limitations, this part of pollution was ignored. Pesticides loss would also cause non-point source pollution. The empirical mode only pesticides use because we did not have pesticide loss data. The non-point source pollution in animal husbandry and pesticides is also worth discussing if there is more wanted data.

## Supporting information

**S1 Fig. Fertilizer input and fertilizer loss in China.**
(DOCX)

**S2 Fig. Parallel trend test.**
(DOCX)

**S3 Fig. Placebo test.** Notes: Panel 1 is the result of the ADL index placebo test; Panel 2 is the result of the placebo test of the number of patients. The X-axis represents the estimated coefficients from 1000 random assignments. The curve is the estimated kernel density distribution. The vertical type is a true estimate in columns (2) and (4) of Table 3.
(DOCX)

 

**S1 File.**
(DOCX)

## Author Contributions

**Conceptualization:** Ying Wang, Chao Chen.

**Formal analysis:** Ying Wang, Hang Xiong.

**Funding acquisition:** Chao Chen.

**Methodology:** Ying Wang, Hang Xiong.

**Supervision:** Chao Chen.

**Writing – original draft:** Ying Wang.

**Writing – review & editing:** Hang Xiong, Chao Chen.

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
