## [Decision Letter · Decision Letter 0]

6 Jan 2022

PONE-D-21-34489Health Effects of Non-point Pollution on the Elderly in Rural Areas of ChinaPLOS ONE

Dear Dr. WANG,

Thank you for submitting your manuscript to PLOS ONE. After careful consideration, we feel that it has merit but does not fully meet PLOS ONE’s publication criteria as it currently stands. Therefore, we invite you to submit a revised version of the manuscript that addresses the points raised during the review process.

You will find that while the reviewers find your paper addressing an important question, they have raised some concerns about the presentation of the paper, the validity of the methods, and the interpretation of the results. Please address their comments point-by-point. 

We look forward to receiving your revised manuscript.

Kind regards,

Yueming Qiu

Academic Editor

PLOS ONE

2. For this observational study, please avoid causal-sounding language (such as 'impact' or 'effect') when reporting associations.

Reviewers' comments:

Reviewer's Responses to Questions

**Comments to the Author**

1. Is the manuscript technically sound, and do the data support the conclusions?

Reviewer #1: Partly

Reviewer #2: Partly

2. Has the statistical analysis been performed appropriately and rigorously? 

Reviewer #1: No

Reviewer #2: No

3. Have the authors made all data underlying the findings in their manuscript fully available?

Reviewer #1: Yes

Reviewer #2: Yes

4. Is the manuscript presented in an intelligible fashion and written in standard English?

Reviewer #1: No

Reviewer #2: Yes

5. Review Comments to the Author

Reviewer #1: This paper focuses on an important and interesting topic of fertilizer exposure/application and human health. Although many efforts are put into this work, the paper still needs some improvement regarding methodology and organization/formatting of the paper. The writing and language should also be improved significantly before being published.

At the beginning of the introduction, a paper usually starts with background information and introduces the issue addressed by the paper. The first paragraph is not clear and interesting enough by listing some findings.

Line 80 The contribution of the paper should be backed up with a thorough literature review. Please add more references and discussion on previous studies that are closely relevant to your study.

Line 98 Can you have a descriptive map showing the provinces and use different color shades to show the levels of fertilizer application/exposure or heath level?

Line 105 What are other regional variables?

Line 152 Why use the average of the past three years instead of using data from the same year? Is there a possibility of lagged effects?

Line 155 I suggest adding min. and max to the table.

Table 1 Provincial variables. Would there be some city-level factors omitted such as city-level subsidies that will influence both fertilizer application and human health?

Line 165 “lines 1 and 2”, do you mean rows?

Line 171 What are the average values of fertilizer loss?

Line 165-line 176 Please give more background information about the agricultural subsidy policy. What it is about and how does it impact fertilizer loss?

Could you have a graph showing how the intensity changes over the years for both the high areas and low areas?

This comparison of high and low is not exactly a difference-in-differences comparison because they both have the policy in place, and they are both “treated”. This is more a two-way fixed effects methodology.

Line 179 Table notes should be self-explanatory. The readers could understand without referring too much to the paper.

Equation (4) There is no subscript for X. Will heavy metals from organic fertilizer also impact human health? Will that bias your estimates?

Line 216 How old is defined as rural elderly and “younger elderly”? Which part of the results is based on equation (4) and which is from equation (5)?

Line 232 Policy implications can be put in the conclusion section.

Table 3 What are the coefficients of the covariates (control variables)? Do they show the expected sign? What are the results without control variables?

Line 248 What do you mean by “reduces the age of the elderly”? Please pay attention to how to interpret the results.

Line 276 How are the costs of fertilizer application compared to the cost of this pesticide use?

Line 301 Do you mean equations (4) and (5)?

Line 317 What are these rice-producing areas. Could you be more specific?

Formatting:

Introduction. Each line ends with a half-word. I suggest checking the composition/distribution of paraphs.

“This is table X legend.” What is this for? Considering removing. Please also check the table adjustment and line spacing.

Grammar mistakes and typos:

There are grammar mistakes and typos. Please also check the uses of some phrases. Examples are seen in the following, but not limited to:

First paragraph in the introduction, “stats, surmise”

line 56 Plication

line 100 “the data of 1998….”

Line 127 “rejuvenation of disease”, “income station

Line 146 should this n be h?

Line 155 Indicaos should be indicators

Line 165 “after the implementation” duplicated phrases

Line 201 medic should be medical

Line 202 than should be then

Some sentences can be cleaned and simplified. I encourage the authors to have some native or fluent speakers read through and help with the language of the paper.

Reviewer #2: Review

This is an important study that covers an important question. Previous studies have shown that fertilizer pollution damage residents’ health and this study intends to show whether the elder residents are more vulnerable to fertilizer pollution. However, there are many major problems with the current manuscript.

1. Although it is important to examine the regional inequality in the medical outcome, the most important question is: why aged people? Previous studies show that fertilizer loss damages residents, then the natural steps should be who are more subjected to harm from fertilizer loss. Therefore, citing a study to show the difference between elder residents and younger residents is appreciated.

2. Therefore, hypothesis 1 needs transition. If fertilizer loss is harmful to health, in general, is known, why do we have to explore its effect on aged residents, before showing whether there is a difference between elder and younger residents? Are they more vulnerable or more resilient than the younger group? This problem can be solved by citing studies showing fertilizer overuse has a different effect on elder and younger people.

3. For hypothesis 2, you might want to add more explanation on how the outcome variables are correlated with the “rejuvenation of disease”. How do you define the term?

4. Hypothesis 3 makes two points, and should be separated; their contents are also made sense by common wisdom, but the literature in the manuscript seems not to correlate with them.

5. The linear regression coefficients are correlations, not “impact”. The impact has a causal implication.

6. This research uses DID, but the research did not check the parallel trends before 2004. In this study, the main analysis uses data after 2004 and separates the provinces into high-loss and low-loss. However, as far as I am aware, there is no mention of how the health outcomes trend before 2004. For DID, parallel trends assumption is essential. I randomly search for a blog from the world bank that could help the authors to have the idea (https://blogs.worldbank.org/impactevaluations/revisiting-difference-differences-parallel-trends-assumption-part-i-pre-trend). Your placebo test does not correlate with this problem.

7. Even if the parallel trend assumption holds, another problem is how to support the assumption that all the residents are treated to high fertilizer overuse randomly. High fertilizer loss could also imply a lack of management or facility, which also leads to high medical loss. For example, it can be argued that the aged person exposed to extra use of fertilizer tend to live in provinces with lower medical resources (agriculture province such as Hubei are also subjected to fertilizer overuse). In short, the author should consider blocking potential causal paths.

8. Another problem that comes with DID is the treatment effect. In formula (4), it seems the treatment effect is continuous. However, the traditional DID does not allow for the continuous treatment effect and could miscalculate S.D and perform the wrong t-test. I suggest reading this paper and reconsidering this part:

Callaway, B., Goodman-Bacon, A., & Sant’Anna, P. H. C. (2021). Difference-in-Differences with a Continuous Treatment. ArXiv:2107.02637 [Econ]. http://arxiv.org/abs/2107.02637

Minor problems are listed as follows:

1. Replacing the word “rejuvenation of disease”. Many definitions can be found on the internet, so it might be helpful to address this word.

2. The research can be more specific on the unit of analysis. “Matches micro-level individual health data and provincial-level chemical fertilizer” is confusing.

3. No need to discuss policy implications in basic regression results of health. I was referring to the line(L)229-L235, L261-257,

4. When you use “1% level is significant”, do you mean “significant at 99% CI”?

5. L316-L319: Are you referring to table 6 or table 7?

6. Results of the placebo test, the Effect of fertilizer input on health, and the Effect in different areas can be moved to the appendix.

To sum up, in its current form, this manuscript is not ready to publish. However, if the author is willing to address these mentioned problems, then the topic is interesting and contributes to the existing literature.

---

## [Author Response · Author response to Decision Letter 0]

14 Apr 2022

Response to Editor

Dear Dr. Qiu,

Thank you for coordinating, reviewing, and extending the invitation to revise our manuscript. We have treated your as well as the reviewers’ comments seriously and have significantly modified portions of the manuscript. For your advice, we have paid specific attention to meeting PLOS ONE’s style and reporting associations in our manuscript. We summarize revisions we have implemented following the two reviewers’ comments. Point by point responses are in separate files addressing to them respectively.

For Reviewer1, we have:

1. Improved writing and readability by simplifying some sentences, proving a better expression, and adding additional reading aid.

2. Added a series of publications in introduction and strengthened our contributions by comparing closely related literatures.

3. Enriched the content of empirical model of by adding regression results.

For Reviewer2, we have:

1. Compared the heterogeneous health risk of environmental exposure by adding the empirical analysis of different age groups and discussion in more detail.

2. Explained the hypotheses more clearly and explained the agricultural support polices in more detail.

3. Revised the description of the regression coefficients. 

4. Offered the parallel trend assumption and discussed consistent estimation of Difference in Differences with a Continuous Treatment.

5. Improved writing and readability by including an additional figure, citing more recent publications, offering better variable definitions, and adding additional reading aid.

We hope you find our revised manuscript acceptable for publication in PLOS ONE.

Sincerely,

Ying Wang

Reviewer: 1 

Thank you for the constructive comments on our manuscript. We have taken your comments seriously and substantially modified portions of the manuscript. We believe our work has been significantly improved as a result of your comments. Please find below our response to each point you made with your original comments in italic. 

This paper focuses on an important and interesting topic of fertilizer exposure/application and human health. Although many efforts are put into this work, the paper still needs some improvement regarding methodology and organization/formatting of the paper. The writing and language should also be improved significantly before being published. 

General comments

1. At the beginning of the introduction, a paper usually starts with background information and introduces the issue addressed by the paper. The first paragraph is not clear and interesting enough by listing some findings.

Response：Following your advice, we have significantly revised the first paragraph. First we introduced the background of agricultural non-point source pollution and fertilizer use in China. Second, added more publications to explain health effect of the agricultural non-point pollution Third, we explained the elders would be more susceptible, and the background of aging in China. This was the reason we focused on the elderly residences in rural China. Last, we have strengthened our contributions by comparing with two works most closely related to our study. The first paragraph was clearer and more logical.

2. Line 80 The contribution of the paper should be backed up with a thorough literature review. Please add more references and discussion on previous studies that are closely relevant to your study.

Response: Thank you very much for your suggestion. We have added some references about agricultural non-point pollution on public health, i.e. pesticide and straw burning in Line41-42, Line52-53. We have cited previous studies relevant to our study to back up the contribution of this paper in Line54-59.

3. Line 98 Can you have a descriptive map showing the provinces and use different color shades to show the levels of fertilizer application/exposure or heath level?

Response: Thank you very much for your suggestion. We dichotomized fertilizer loss by its pre-policy mean. The national average value of fertilizer loss is 5.36 kg/ha. The red parts in Figure 1 represent regions with high intensity of fertilizer loss and the pink parts represent are for regions with low intensity of fertilizer loss. 

Fig 1 Fertilizer Loss in China

4. Line 105 What are other regional variables?

Response: Thank you very much for your suggestion. The other regional variables contain fertilizer use, organic fertilizer use, pesticide use, industrial pollution index, numbers of hospital and provincial Gross Domestic Product. GDP measures general economy and wealth conditions. The number of hospital controls for the level of medical service. Industrial pollutions control for pollution from industrial activities. All the regional variables are described in section “Provincial variables” in Section 2.1.3 “Provincial variables”.

5. Line 152 Why use the average of the past three years instead of using data from the same year? Is there a possibility of lagged effects?

Response: The consideration is as following. First, The individual health data from the Chinese Longitudinal Healthy Longevity Survey. We observed were once every three years. Second, as mentioned in our article, most studies indicated a lagged effects of environmental exposure on public health(e.g. Wang and Luo 2021; Hu et al. 2015) Therefore, we follow the literature and also use the average of the past three year considering the lagged effect as well as the consistency of health data.

6. Line 155 I suggest adding min. and max to the table.

Response: Thank you for the advice. We have added the min. and max in Table1.

7. Table 1 Provincial variables. Would there be some city-level factors omitted such as city-level subsidies that will influence both fertilizer application and human health?

Response: Thank you very much for your suggestion. This study explores the health effect of fertilizer loss at the provincial level and we have controlled the individual effects, so we did not control the city-level factors. We did not control the city-level subsidies because the samples are rural residents, not only the farmers who receive the health damage via direct exposure. We hold the fertilizer loss is exogenous. 

8. Line 165 “Lines 1 and 2”, do you mean rows?

Response: Thank you for the suggestion. “Line1 and 2” means the rows of ADL index and No. of diseases, respectively. “Rows or Lines” maybe nor clearly enough, so we reinterpret the results of Table 2 in Section 2.2.2. 

9. Line 171 What are the average values of fertilizer loss?

Response: The fertilizer loss is continuous variable, and we dichotomized fertilizer loss by its pre-policy mean to compare the health effect in different regions. The average value of fertilizer loss is the national average of fertilizer loss; it is equal to 5.36kg/ha. We explained the details in section “Supporting information”.

10. Line 165-Line 176 Please give more background information about the agricultural subsidy policy. What it is about and how does it impact fertilizer loss?

Response: Thank you for the suggestion. We have added the background about agricultural subsidy policy and the effect of fertilizer loss on environment and public health in Section 2.2.2.

11. Could you have a graph showing how the intensity changes over the years for both the high areas and low areas?

Response: Thank you for the suggestion. The fertilizer loss intensity and intensity variations are shown in Fig 2. Fertilizer loss increase per hectare in high loss regions is roughly 2times larger than low loss areas. The variations of fertilizer input intensity, fertilizer loss and fertilizer loss intensity were showed in Table2 in the revised manuscript.

Fig 2 Fertilizer Loss Amount and Variation 

12. This comparison of high and low is not exactly a difference-in-differences comparison because they both have the policy in place, and they are both “treated”. This is more a two-way fixed effects methodology.

Response: Thank you for the suggestion. As you pointed out, all the samples are treated by agricultural support policies. A difference-in-differences is still applicable, but with two conditions need to be satisfied. First, the impacts are different between “experience group” and “control group”. Second, the parallel trend assumption is need be hold. Following publications adopted DID to explain the “policy effect”, however all samples were treated by the “policy”: the bill of strengthening the protection of mortgage claims will have great impact on enterprises with high proportion of tangible assets, because the tangible assets are easier to be used as collateral, so Vig(2013) constructed the experimental group and the control group according to the proportion of tangible assets of enterprises. Campello &Larrain (2016) constructed the experimental group and control group according to the difference of enterprises' demand for movable property. In our study, we constructed the experimental group and control group by fertilizer loss levels. First, the parallel trend assumption holds. Second, most of the high loss areas are in the developed regions and is the primary rice production regions in China. Agricultural subsidies of per unit of field are different between provinces, such as the subsidies in Jilin province is four and two times lower than Guangdong and Zhejiang respectively (Shu et al.,2013). The developed provinces had a larger subsides, and this will cause a larger fertilizer loss after agricultural support policies. The Green Revolution lead to the extensive use of chemical fertilizer, particular in rice production. We dichotomized fertilizer loss by its pre-policy mean. The variation of fertilizer loss in high loss areas is 2 times larger than low loss areas. Also, we used a continuous measure of intensity of treatment (i.e. fertilizer loss) and thereby captured more variation in the data (Nunn and Qian, 2011; Lai, 2017). As your suggestion, we have estimate the health effect of fertilizer loss using TWFE, and we have added the estimated results in column (3) and column (6) of Table3.

13. Line 179 Table notes should be self-explanatory. The readers could understand without referring too much to the paper.

Response: Thank you for your advice. We have deleted the table notes. 

14. Equation (4) There is no subscript for X. Will heavy metals from organic fertilizer also impact human health? Will that bias your estimates?

Response: Thank you for the suggestion. We have added the subscript for X in Equation (4). Chemical fertilizer was most widely used in China and the supply proportion of chemical fertilizer nutrients exceeds seventy percent. The use of organic fertilizer was far less than that of chemical fertilizer, and the organic fertilizer use was concentrated in some regions. So we did not consider the health effect of organic fertilizer. But following your suggestion, we have controlled the organic fertilizer use in our revised manuscript. 

15. Line 216 How old is defined as rural elderly and “younger elderly”? Which part of the results is based on equation (4) and which is from equation (5)?

Response: Thank you for the suggestion. According to the previous studies, an elderly person over 80 years or 85 years is defined as the oldest olds. The oldest olds were wildly discussed because they had greater risk of falling and hospitalization (Rockwood et al. 1999; Speechley &Tinetti, 1991; Winograd,1991; Brettschneider et al. 2019). We divided the samples by age into the younger elderly samples (<80years) and the oldest olds samples (≥80 years). The coefficient of equation (4) represented the post-agricultural policies of fertilizer loss. health effect of fertilizer loss and the of equation (5) captured the post-agricultural policies effect of high loss areas relative to low loss areas. Estimated coefficients of were reported in column (1) and column (3) of Table3, and estimated coefficients of were reported in column (2) and column (4) of Table3. We described the result of Table 3 in section 3.1.1.

16. Line 232 Policy implications can be put in the conclusion section.

Response: Thank you for this advice. We have deleted the policy implications Line229-235, Line261-257 of original manuscript. And we discussed the policy implications in conclusion section.

17. Table 3 What are the coefficients of the covariates (control variables)? Do they show the expected sign? What are the results without control variables?

Response: We have added the results without control variables in S1 Table in section “Support information” in our revise manuscript. The results with covariates are shown in Table 1 below. 

Table 1 Basic Estimates ADL index and Diseases

 ADL No. of diseases

 (1) (3) (2) (4)

Fertilizerloss*Time 0.0147** 0.0057*** 

 (0.0061) (0.0014) 

Higharea*Time 0.1222*** 0.0487***

 (0.0474) (0.0116)

Floss 0.0570 0.0659 -0.0148 -0.0124

 (0.0417) (0.0416) (0.0103) (0.0103)

Pinpur -94.2811 -125.0235 -47.6648 -65.0932**

 (116.5155) (118.8320) (30.3544) (31.1875)

OFinput 0.0010 0.0010 0.0009*** 0.0009***

 (0.0010) (0.0010) (0.0002) (0.0002)

Age 0.0273 0.0323 0.0113 0.0109

 (0.0318) (0.0317) (0.0081) (0.0081)

Male -0.0103 -0.0373 -0.0029 -0.0042

 (0.2373) (0.2385) (0.0426) (0.0427)

Co-residence 0.1617*** 0.1647*** -0.0074 -0.0077

 (0.0256) (0.0256) (0.0066) (0.0066)

Income_cost 0.0354* 0.0271 -0.0024 -0.0023

 (0.0200) (0.0199) (0.0046) (0.0046)

Illness 1.5128*** 1.4911*** 3.5079*** 3.5068***

 (0.1930) (0.1923) (0.0580) (0.0581)

Hospitalization 0.1171*** 0.1170*** 0.0126*** 0.0126***

 (0.0183) (0.0183) (0.0038) (0.0038)

Smoke -0.0784 -0.0722 0.0369* 0.0366*

 (0.0856) (0.0851) (0.0201) (0.0201)

Drink -0.0330 -0.0264 -0.0104 -0.0103

 (0.0663) (0.0662) (0.0167) (0.0167)

Exercise -0.2192*** -0.2181*** -0.0123 -0.0123

 (0.0489) (0.0488) (0.0127) (0.0127)

Meat 0.0196 0.0044 0.0056 0.0054

 (0.0140) (0.0184) (0.0044) (0.0044)

Fish 0.0245* 0.0132 -0.0014 -0.0015

 (0.0131) (0.0184) (0.0045) (0.0045)

Egg -0.0140 -0.0247 0.0024 0.0025

 (0.0116) (0.0183) (0.0046) (0.0046)

Salt_veg -0.0435*** -0.0406*** -0.0075*** -0.0075***

 (0.0089) (0.0089) (0.0024) (0.0024)

Boild water -0.0836* -0.0934** -0.0042 -0.0050

 (0.0456) (0.0457) (0.0107) (0.0107)

Water 0.0028 0.0027 -0.0152 -0.0147

 (0.0823) (0.0822) (0.0175) (0.0175)

Hospnum 0.0712*** 0.0706*** -0.0155*** -0.0159***

 (0.0153) (0.0154) (0.0041) (0.0041)

LnGDP -0.2306 -0.2379 0.0920 0.0996*

 (0.2207) (0.2185) (0.0585) (0.0576)

Indpolltion 0.0662 0.0495 0.0291 0.0279

 (0.1085) (0.1081) (0.0253) (0.0253)

Constant 4.8844 5.0608 -1.6139* -1.6166*

 (3.3262) (3.3242) (0.8570) (0.8543)

Observations 32,165 32,165 32,329 32,329

R-squared 0.5570 0.5582 0.6906 0.6907

Year FE YES YES YES YES

ID FE YES YES YES YES

robust standard errors for clustering at the individual level are reported in brackets; *, **, *** are significant at the level of 1%, 5% and 10% respectively

18. Line 248 What do you mean by “reduces the age of the elderly”? Please pay attention to how to interpret the results.

Response: Thank you for the advice. “Reduces the age of elderly” means the fertilizer loss will lead the age of illness onset younger. As your suggestions, we have deleted this confusing expression and interpreted the results of Table 4 in section 3.1.3.

19. Line 276 How are the costs of fertilizer application compared to the cost of this pesticide use?

Response: Thank you for the suggestion. This study focused on the health effect and medical cost loss of fertilizer. We have estimated the medical cost of fertilizer loss. An increase of one unit (1kg/ha) of fertilizer loss would add CNY 3.74 (USD 0.53) to medical cost, which would increase 316 million yuan (45 million dollars) in total for the rural elderly populations in China. For you advise, we added medical cost of fertilizer input in Line290-293. One more unit fertilizer input will increase medical cost by CNY 0.22. For the elderly population in rural China, the total medical cost will increase CNY 21.5 (USD 3.1) million.

20. Line 301 Do you mean equations (4) and (5)?

Response: Sorry about these errors. We have corrected these errors.

21. Line 317 What are these rice-producing areas. Could you be more specific?

Response: Thank you for the suggestion. The details about major agricultural-producing areas, major rice-producing areas and water resource-rich areas are in section supporting information.

22. Formatting:

Introduction. Each Line ends with a half-word. I suggest checking the composition/distribution of paraphs.

Response: Thank you for this suggestion. We have revised it in introduction and other parts of our manuscript.

23. “This is table X legend.” What is this for? Considering removing. Please also check the table adjustment and Line spacing.

Response: Thank you for the suggestion. “This is table X legend” is the template format. We have adjusted the single spacing of the tables.

24. Grammar mistakes and typos:

There are grammar mistakes and typos. Please also check the uses of some phrases. Examples are seen in the following, but not limited to:

First paragraph in the introduction, “stats, surmise”

Line 56 Plication

Line 100 “the data of 1998….”

Line 127 “rejuvenation of disease”, “income station

Line 146 should this n be h?

Line 155 Indicaos should be indicators

Line 165 “after the implementation” duplicated phrases

Line 201 medic should be medical

Line 202 than should be then

Some sentences can be cleaned and simplified. I encourage the authors to have some native or fluent speakers read through and help with the language of the paper.

Response: Sorry for these errors. In our revise manuscript, we have carefully checked the grammar and revised the sentences. 

Reviewer: 2

Thank you for the constructive comments on our manuscript. We have taken your comments seriously and substantially modified portions of the manuscript. We believe our work has been significantly improved as a result of your comments. Please find below our response to each point you made with your original comments in italic.

This paper focuses on an important and interesting topic of fertilizer exposure/application and human health. Although many efforts are put into this work, the paper still needs some improvement regarding methodology and organization/formatting of the paper. The writing and language should also be improved significantly before being published. At the beginning of the introduction, a paper usually starts with background information and introduces the issue addressed by the paper. The first paragraph is not clear and interesting enough by listing some findings.

General comments

1. Although it is important to examine the regional inequality in the medical outcome, the most important question is: why aged people? Previous studies show that fertilizer loss damages residents, then the natural steps should be who are more subjected to harm from fertilizer loss. Therefore, citing a study to show the difference between elder residents and younger residents is appreciated.

Response：Thank you for the suggestion. China has the most serious aging, 70% of elderly live in rural areas. Agricultural non-point pollutions had been more serious according to the overuse of agrochemicals. Under the background of aging and non-point pollution, this study focused on health effect of fertilizer loss on elderly in rural areas. The elder residents have different vulnerabilities to environmental contaminants. We have cited several studies in Line 44-48.

2. Therefore, hypothesis 1 needs transition. If fertilizer loss is harmful to health, in general, is known, why do we have to explore its effect on aged residents, before showing whether there is a difference between elder and younger residents? Are they more vulnerable or more resilient than the younger group? This problem can be solved by citing studies showing fertilizer overuse has a different effect on elder and younger people.

Response: Several studies had reported that the elder residents were more vulnerable than the younger groups. A higher susceptibility and higher mortality rates could be observed among the elder residents (We cited the studies in Introduction(Line44-45). We have added more references to explain the reasons of heterogeneity effect in younger and older population in Line 45-48. We further explored heterogeneous health effect of fertilizer loss by splitting the samples in various ages. First, we divided the samples at age into the younger age samples (<80years) and the oldest olds samples (≥80 years). Then, we also divided the samples into younger olds (65years-75years), middle olds (75-85years) and oldest olds (≥85years) according to Monti et al. 2021 and Lau et al. 2021. The heterogeneous effects are in section 3.1.2.

3. For hypothesis 2, you might want to add more explanation on how the outcome variables are correlated with the “rejuvenation of disease”. How do you define the term?

Response: We have rewired the hypothesis 2. Because of the physical function, fertilizer loss may lead the samples to be ill younger. The fertilizer loss caused the several diseases, which in Line76-83. And in Figure 2, the elderly under 80 years of age, there was a significant difference in the age of illness onset in the areas with high chemical fertilizer loss, and the age of illness onset in the elderly in the areas with high chemical fertilizer loss was lower than that in the areas with low chemical fertilizer loss.

4. Hypothesis 3 makes two points, and should be separated; their contents are also made sense by common wisdom, but the literature in the manuscript seems not to correlate with them. 

Response: Thank you for the suggestion. We revised the hypothesis 3 by “Due to the health damages, the medical cost will be higher in high-loss regions comparing the regions with lower fertilizer loss”. In this section, we explored the economic results by translating estimated effects into expected monetary loss.

5. The linear regression coefficients are correlations, not “impact”. The impact has a causal implication.

Response: Thank you for the suggestion. We have revised the expressions by using “lead to, be associated with”.

6. This research uses DID, but the research did not check the parallel trends before 2004. In this study, the main analysis uses data after 2004 and separates the provinces into high-loss and low-loss. However, as far as I am aware, there is no mention of how the health outcomes trend before 2004. For DID, parallel trends assumption is essential. I randomly search for a blog from the world bank that could help the authors to have the idea (https://blogs.worldbank.org/impactevaluations/revisiting-difference-differences-parallel-trends-assumption-part-i-pre-trend). Your placebo test does not correlate with this problem.

Response: Thank you for the suggestion. We added the parallel trends in section 3.3.1. We divided the samples to six sections: the second period before the policy implementation, the first period before the policy implementation, the second period after policy, the second two periods after the policy implementation to the end of sample period. Thus, six indicator variables pre_2, pre_1, post_1, post_2, post_3 and post_4 were constructed. The set of pre_1 coefficient were statistically indistinguishable from zero, but there was a sharp, statistically significant and sustained change after the policy implementation.

7. Even if the parallel trend assumption holds, another problem is how to support the assumption that all the residents are treated to high fertilizer overuse randomly. High fertilizer loss could also imply a lack of management or facility, which also leads to high medical loss. For example, it can be argued that the aged person exposed to extra use of fertilizer tend to live in provinces with lower medical resources (agriculture province such as Hubei are also subjected to fertilizer overuse). In short, the author should consider blocking potential causal paths.

Response: Thank you for the suggestion. We had considered the randomization of fertilizer loss. A major concern was that the factors that caused the different fertilizer loss between provinces may be correlated with health outcomes. These types of confounding factors vary across provinces and individuals but are fixed over time. A common method of controlling for time-invariant unobserved heterogeneity is to use fixed effects. We had fixed the individual effect and time effect. And we also had controlled the medical resources and economic developments. 

8. Another problem that comes with DID is the treatment effect. In formula (4), it seems the treatment effect is continuous. However, the traditional DID does not allow for the continuous treatment effect and could miscalculate S.D and perform the wrong t-test. I suggest reading this paper and reconsidering this part:

Callaway, B., Goodman-Bacon, A., & Sant’Anna, P. H. C. (2021). Difference-in-Differences with a Continuous Treatment. ArXiv:2107.02637 [Econ]. http://arxiv.org/abs/2107.02637

Response: Thank you for this advice and thank you very much for recommending the paper. We have learned the paper you recommended carefully, the paper provided a good job in the front of theoretical basis research of Difference-in-Differences. Unfortunately to date a package that allows researchers to implement this approach in Stata has not been released. And sorry for that we are not capable of empirical research following Callaway’s theory. However, if differences in treatment intensity are not correlated with heterogeneity in the treatment effect at a given treatment intensity or there is no selection of outcome variables, the traditional DiD can be used (Callagher, et al. 2021; Kindsgrab, 2022; Battiston et al.2021). Callaway argued that the magnitude of coefficient with continuous treatment from DiDs should be interpreted with caution as the interaction coefficient identifies a weighted average of the “average causal response” of the treatment along different levels of the treatment. In our setting, we are interested in causal parameter the average treatment effect to fertilizer loss level . When we want to compare the magnitudes of across different levels of health damage (i.e. Equation4 and Equation6), we need a stronger parallel trends assumption that the restricts the potential outcomes of individuals. Specifically, we must assume that the average potential outcomes for individuals are the same at each level of fertilizer loss, or that (on average) there is no selection into a particular level of fertilizer loss. The stronger parallel trends assumption is likely to be satisfied in our setting. In addition, we followed some literatures by adopting a binary treatment as robustness check (Gong et al.2022; Motohashi,2022). 

Minor problems are listed as follows:

9. Replacing the word “rejuvenation of disease”. Many definitions can be found on the internet, so it might be helpful to address this word.

Response: Thank you for thesuggestion. In this study, “rejuvenation of disease” means that the residences may be ill in younger ages. We have changed the expression by “be ill younger” and “be sick in younger age”.

10. The research can be more specific on the unit of analysis. “Matches micro-level individual health data and provincial-level chemical fertilizer” is confusing.

Response: Thank you for the advice. The analysis was accomplished using difference in differences method with the national dataset Chinese Longitudinal Healthy Longevity Survey (CLHLS) and publicly published yearbooks. We have revised the explanation in first paragraph of Section 2.1 “Data Collection and Variables”. We have introduced the CLHLS database and how did the individual variables defined in section 2.1.1 and section 2.1.2. We also explained the provincial variables in section 2.1.3. 

11. No need to discuss policy implications in basic regression results of health. I was referring to the line(L)229-L235, L261-257,

Response: Thank you for the suggestion. We have deleted the discussion of policy implications.

12. When you use “1% level is significant”, do you mean “significant at 99% CI”?

Response: Thank you for the suggestion. Yes, “1% level is significant” means “significant at 99% CI”. “1% level is significant” is usually used in economics.

13. L316-L319: Are you referring to table 6 or table 7? 

Response: Sorry for these errors. Line316-319 is the descriptions of Table 7. And we have moved Table7 to section “Supporting information”, in S3 Table and S4 Table.

14. Results of the placebo test, the Effect of fertilizer input on health, and the Effect in different areas can be moved to the appendix.

Response: Thank you for the suggestion. We have moved the robustness check to the Supporting information.

To sum up, in its current form, this manuscript is not ready to publish. However, if the author is willing to address these mentioned problems, then the topic is interesting and contributes to the existing literature.

Reference

1. Battiston D, Espinosa M, Liu S. Talent poaching and job rotation. SSRN. 2021; 3778068,

2. Brettschneider C, Hajek A, Rohr S, Fuchs A, Weeg D, et al. Determinants of health-care costs in the oldest-old in Germany. The Journal of the Economics of Ageing. 2019; 14: 100200.

3. Campello M, Larrain M. Enlarging the contracting space: Collateral menus, access to credit, and economic activity. The Review of Financial Studies. 2016; 29(2): 349-383.

4. Gallagher J, Hartley D, Rohlin S. Weathering an unexpected financial shock: the role of federal disaster assistance on household finance and business survival. 2021.

5. Gong D, Yan A, Yu J. Cost of zero-covid: effects of anti-contagious policy on labor market outcomes in china[J]. SSRN.2022; 4037688.

6. He GJ, Liu T, Zhou MG. Straw burning, PM 2.5, and death: evidence from China. Journal of Development Economics. 2020;145: 102468.

7. Hu R, Huang X, Huang J, Li Y, Zhang C, et al. Long- and short-term health effects of pesticide exposure: a cohort study from China. PLOS One. 2015; 10(6): e0128766. 

8. Kindsgrab P. Do higher income taxes on top earners trickle down? A local labor markets approach. A Local Labor Markets Approach, 2022.

9. Lai WY. Pesticide use and health outcomes: evidence from agricultural water pollution in China. Journal of Environmental Economics and Management. 2017; 86: 93-120. 

10. Motohashi K. Unintended consequences of sanitation: externalities on water quality and health in India.2022.

11. Nunn N, Qian N, The potato's contribution to population and urbanization: evidence from a historical experiment. The Quarterly Journal of Economics. 2011;126(2): 593-650.

12. Rockwood K, Stadnyk K, MacKnight C, McDowell I. A brief clinical instrument to classify frailty in elderly people. The Lancet. 1999; 353:205-206.

13. Shu KL, Yang S, Wang HL, Guo QH. Analysis on evolution of Jilin grain production investment mechanism and its relevant policy effect. Research of Agricultural Modernization. 2013;34(03): 257-262. 

14. Speechley M, Tinetti M. Falls and injuries in frail and vigorous community elderly persons. Journal of the American Geriatrics Society. 1991; 39:46-52. 

15. Wang Y Z, Luo N S. Air pollution, health depreciation and medical costs: Research based on the three perspectives of physical health, mental health and social adaptability[J]. Economic Research Journal, 2020, 55(12): 80-97.

16. Winograd CH. Targeting Strategies: An overview of criteria and outcomes. Journal of the American Geriatrics Society. 1991; 39:25-35. 

17. Vig V. Access to collateral and corporate debt structure: evidence from a natural experiment. The Journal of Finance. 2013; 68(3): 881-928.

---

## [Decision Letter · Decision Letter 1]

26 May 2022

PONE-D-21-34489R1Agricultural Nonpoint-source Pollution and Health of the Elderly in Rural ChinaPLOS ONE

Dear Dr. WANG,

Thank you for submitting your manuscript to PLOS ONE. After careful consideration, we feel that it has merit but does not fully meet PLOS ONE’s publication criteria as it currently stands. Therefore, we invite you to submit a revised version of the manuscript that addresses the points raised during the review process.

We look forward to receiving your revised manuscript.

Kind regards,

Yueming Qiu

Academic Editor

PLOS ONE

Journal Requirements:

Reviewers' comments:

Reviewer's Responses to Questions

**Comments to the Author**

1. If the authors have adequately addressed your comments raised in a previous round of review and you feel that this manuscript is now acceptable for publication, you may indicate that here to bypass the “Comments to the Author” section, enter your conflict of interest statement in the “Confidential to Editor” section, and submit your "Accept" recommendation.

Reviewer #1: All comments have been addressed

Reviewer #2: All comments have been addressed

2. Is the manuscript technically sound, and do the data support the conclusions?

Reviewer #1: Yes

Reviewer #2: Partly

3. Has the statistical analysis been performed appropriately and rigorously? 

Reviewer #1: Yes

Reviewer #2: No

4. Have the authors made all data underlying the findings in their manuscript fully available?

Reviewer #1: Yes

Reviewer #2: Yes

5. Is the manuscript presented in an intelligible fashion and written in standard English?

Reviewer #1: No

Reviewer #2: Yes

6. Review Comments to the Author

Reviewer #1: The manuscript is much improved. I have the following comments before this paper can be published. I have also highlighted some minor comments such as grammar mistakes, typos, and formatting errors in the revised manuscript (see pdf if applicable).

Abstract: “ADL daily life index of 0.0147..” How much is this change in percentage?

Figure 1: Can you add something about the pathways, such as through drinking water?

I did not see a figure showing the Fertilizer loss in China (maybe you can put it into the Supporting information)

Figure 2: Could you add a vertical line to show treatment time?

Table 3 Missing variable name for coefficients. Please add. Also, could you check the R2 for column (3)?

What are the model specifications for TWFE?

Table 4 Do individual fixed effects mean the individual person (survey participants) fixed effects or province fixed effects?

Section 3.2.1 “The differences between younger and middle olds are not significant in both ADL and No. of diseases”. Do you expect that the effects are homogenous across ages? Are there any possible explanations if they are the same?

Table 5 what are the p values?

Section 3.3.1 Parallel trend test Why are they divided into five periods?

Table S3 What do that superscripts for p values mean?

Reviewer #2: 6. Your parallel trend analysis is appreciated. However, you might want to add an explanation on which figures are for low fertilizer regions and which are for the high fertilizer regions. Otherwise, I cannot understand. Also, the explanation "We divide the samples to five sections" is also confusing. You could consider rewriting this part.

7. I am concerned about this treatment. Although in some cases, adding linear coefficients could work, health factors are sometimes not linear. Do the authors have considered matching? If so, how does it go?

8. I know it makes our life difficult if no package is available. However, assuming the level of fertilizer loss does not have different impacts is too strong. After all, this is the main argument of this study. I would suggest, that given the authors already adopt binary treatment in the robustness check, the authors should repeat the binary robustness check by calibrating the cutoff for the binary treatment effect.

7. PLOS authors have the option to publish the peer review history of their article (what does this mean?). If published, this will include your full peer review and any attached files.

Reviewer #1: No

Reviewer #2: **Yes: **Arthur Lin Ku

---

## [Author Response · Author response to Decision Letter 1]

5 Jul 2022

Response to Editor

Dear Dr. Qiu,

Thank you for coordinating, reviewing, and extending the invitation to revise our manuscript. We have treated yours as well as the reviewers’ comments seriously and made significant modification to the manuscript. As follows, we will summarize revisions implemented following the two reviewers’ comments before point by point responses.

For Reviewer1, we have:

1. Improved writing and readability by proving a better expression and revising Figure 1 and Figure 2.

2. Added the regression results of percentage changes and the details of TWFE model.

For Reviewer2, we have:

1. Revised the section of “parallel trend analysis” and explained the regression results in more details.

2. Added the regression results of propensity score matching (PSM) difference-in-differences (DID).

3. Offered the binary robustness check by changing the cutoff for the binary treatment effect. 

We hope you find our revised manuscript acceptable for publication in PLOS ONE.

Sincerely,

Ying Wang

Cc. Hang Xiong and Chao Chen

 

Reviewer: 1 

Thank you for the constructive comments on our manuscript. We have taken your comments seriously and substantially modified the manuscript. We believe our work has been significantly improved thanks to your comments. Please find below our response to each point you made with your original comments in italic. 

The manuscript is much improved. I have the following comments before this paper can be published. I have also highlighted some minor comments such as grammar mistakes, typos, and formatting errors in the revised manuscript (see pdf if applicable).

1.Abstract: “ADL daily life index of 0.0147..” How much is this change in percentage? 

Response：Thank you very much for your question. ADL daily index of 0.0147 is 0.2 percent change. We have added the percentages in the abstract. The empirical model results are in Table1. The explained variables are the logarithmic of health outcome variables.

Table 1 Empirical result of the logarithmic of health outcome variables.

 ADL No. of diseases

 (1) (2) (3) (4)

Fertilizerloss*Time 0.0020*** 0.0017** 0.0024 0.0047*

 (0.0006) (0.0007) (0.0030) (0.0028)

Control NO YES NO YES

Year FE YES YES YES YES

ID FE YES YES YES YES

Observations 32189 32165 4426 4395

R-squared 0.5618 0.5719 0.5245 0.6093

Notes: 

Robust standard errors for clustering at the individual level are reported in brackets.

*, **, *** are significant at the level of 10%, 5% and 1% respectively.

2.Figure 1: Can you add something about the pathways, such as through drinking water?

Response：Following your advice, we have revised the Figure 1.

3.I did not see a figure showing the Fertilizer loss in China (maybe you can put it into the Supporting information)

Response：Thank you very much for your suggestion. We have added S1 figure to demonstrate the Fertilizer Input and Fertilizer Loss in China in the Supporting information.

 4.Figure 2: Could you add a vertical line to show treatment time? 

Response：Following your advice, we have added a vertical line to show treatment line and we have replaced the figure2.

5.Table 3 Missing variable name for coefficients. Please add. Also, could you check the R2 for column (3)? 

Response：Sorry about these errors. In our revised manuscript, we have carefully corrects these errors.

6.What are the model specifications for TWFE?

Response：Thanks for your question. We use two-way fixed effect model to estimate the health effect of fertilizer loss directly. We fix the individual and time effects both.

 (1)

where demotes the health outcome; identifies the marginal effect of fertilizer loss; is a vector of control variables including individual, household, and provincial characteristics such as age, gender, health behavior, dietary habits, industrial pollution and hospitals, etc.; is a fixed effect unique to individuals ; is a time effect common to all individuals in year ; and is an error term.

7.Table 4 Do individual fixed effects mean the individual person (survey participants) fixed effects or province fixed effects?

Response：In this paper, individual fixed effects mean the individual person (survey participants) fixed effects. And we have revised by “ID FE” to make the expression clearer. 

8.Section 3.2.1 “The differences between younger and middle olds are not significant in both ADL and No. of diseases”. Do you expect that the effects are homogenous across ages? Are there any possible explanations if they are the same? 

Response：Thank you for the questions. The health effects of pollution expose are complex and does not follow a simple linear or near-linear relationship. The elderly often has a higher susceptibility and a higher mortality rate (Wong et al., 2015; Cohen and Gerber, 2017). The heterogeneous effects are because of the age-related behaviors and the physical functions (Tuttle et al., 2013; Rockwood et al., 1999). Increasing variability in diseases and disabilities could be found when people aged and the aging people were more likely to fall and be hospitalized (Speechley and Tinetti, 1991; Winograd, 1991). The younger elderly had better physical functions, so effects on ADL and No. of diseases are not significant. However, the younger elderly also has health risks. The elderly becomes suffered from the diseases related to fertilizer loss as the age growing. So a significant effect on No. of diseases could be observed among the middle olds elderly. 

9.Table 5 what are the p values?

Response：P-values estimate the likelihood of obtaining the observed differences in coefficient estimates. The original hypothesis is there is no difference between groups ( ).The model results reject the original hypothesis. It means that there is a significant difference between groups.

10.Section 3.3.1 Parallel trend test Why are they divided into five periods? 

Response：Thank you very much for your question. There are six waves (2000, 2002, 2005, 2008, 2011, 2014) survey data. Two periods before the agricultural support policies, and four periods after treatment time. We choose the first period before the treatment time point as the benchmark group, so there are five periods. We have restructured this part.

11.Table S3 What do that superscripts for p values mean?

Response：P-values estimate the likelihood of obtaining the observed differences in coefficient estimates.

Response for comments in the PDF documents:

1. Thank you very much for the suggestion about typographical and grammatical errors. We have corrected the errors. And we have checked publications in the journal and reconfirmed that the supporting information citations as “S1 Fig” and “S1 Table”.

2. We have revised the section of conclusions and discussion and shortened the expression of Line344-347.

3. For response 9: We have added the details about the average values of fertilizer loss in note of Table 2 in our revised manuscript.

4. For response 17: There is a double-direction causality relationship between medical supply and health. And the medical supply is only a control variable, so we did not discuss the double-direction causality relationship. Further, we estimated health effect of fertilizer loss by deleting the medial supply variable (the results in Table 2). Also the negative effects between medical supply and health were found in other literatures (Lai, 2017; Ju,2022), when they controlled medical supply.

Table 2 Estimates without variable of medical supply

 ADL No. of diseases

Fertilizerloss*Time 0.0173*** 0.0051***

 (0.0061) (0.0014)

Control YES YES

Year FE YES YES

ID FE YES YES

Observations 32165 32329

R-squared 0.5564 0.6903

Notes:

*, **, *** are significant at the level of 10%, 5% and 1% respectively.

 

Reviewer: 2 

 6. Your parallel trend analysis is appreciated. However, you might want to add an explanation on which figures are for low fertilizer regions and which are for the high fertilizer regions. Otherwise, I cannot understand. Also, the explanation "We divide the samples to five sections" is also confusing. You could consider rewriting this part.

Response：Thank you very much for your suggestion. In order to test whether the time staggered entry events of agricultural policy are effective exogenous shocks, we conduct a parallel test base on Equation (9):

 (9)

where is a binary variable that takes value of 1 when agricultural support policies are implemented. We use to denote the base year. Panel A and Panel B of S2 Fig were the estimate results of fertilizer loss and Panel C and Panel showed the estimated coefficients when adopting a binary variable that takes values 1 as high loss areas. Following your advice, we have restructured the Section of Parallel trend test. We added the Equation (9) to explain how we do the parallel trend, and we explained S2 Fig in more detail.

7. I am concerned about this treatment. Although in some cases, adding linear coefficients could work, health factors are sometimes not linear. Do the authors have considered matching? If so, how does it go?

Response：Following your advice, three matching techniques are utilized to assess the health effect of fertilizer loss, namely nearest neighbor matching, the radius matching and the kernel matching. The matching results are reported in Table 3. The estimated coefficients are significant and it means that the basic results are robust. We have added the Table 3 in support information as S2 Table in our revised manuscript.

Table 3 Estimates of Matching

 Nearest neighbor Radius Kernel

 ADL No. of diseases ADL No. of diseases ADL No. of diseases 

Fertilizerloss*Time 0.0695** 0.0143** 0.0267*** 0.0065*** 0.0267*** 0.0065***

 (0.0273) (0.0064) (0.0085) (0.0023) (0.0085) (0.0023)

Control YES YES YES YES YES YES

Year FE YES YES YES YES YES YES

ID FE YES YES YES YES YES YES

Observations 4532 4532 17801 17801 17801 17801

R-squared 0.5927 0.7152 0.5828 0.7049 0.5828 0.7049

a *, **, *** are significant at the level of 10%, 5% and 1% respectively.

8. I know it makes our life difficult if no package is available. However, assuming the level of fertilizer loss does not have different impacts is too strong. After all, this is the main argument of this study. I would suggest, that given the authors already adopt binary treatment in the robustness check, the authors should repeat the binary robustness check by calibrating the cutoff for the binary treatment effect.

Response：Following your advice, we have offered the binary robustness check by changing the cutoff for the binary treatment effect. We defined the high loss areas with the top 10%, 20%, 30% and 40% of the highest fertilizer loss intensity respectively. Table 4 reports the estimate coefficients in Equation (5), the basic regression results of binary treatment are robust to a variety of robustness checks.

Table 4 The Robustness Checks of Binary Treatment

 ADL No. of diseases

 10% 20% 30% 40% 10% 20% 30% 40%

Higharea*Time 0.1193** 0.1278** 0.1946*** 0.1890*** 0.0239* 0.0621*** 0.0500*** 0.0671***

 (0.0596) (0.0525) (0.0500) (0.0491) (0.0142) (0.0122) (0.0119) (0.0120)

Control YES YES YES YES YES YES YES YES

Year FE YES YES YES YES YES YES YES YES

ID FE YES YES YES YES YES YES YES YES

Observations 32,189 32,189 32,189 32,189 32,353 32,353 32,353 32,353

R-squared 0.5620 0.5621 0.5623 0.5623 0.6898 0.6901 0.6900 0.6902

a robust standard error for clustering at the individual level are reported in brackets; *, **, *** are significant at the level of 10%, 5% and 1% respectively 

Reference

1. Wong CM, Lai HK, Tsang H, et al. Satellite-based estimates of long-term exposure to fine particles and association with mortality in elderly Hong Kong residents. Environmental Health Perspectives. 2015;123(11):1167-1172

2. Cohen, G., Gerber, Y. Air pollution and successful aging: Recent evidence and new perspectives. Current Environmental Health Reports. 2017; 4:1-11. https://doi.org/10.1007/s40572-017-0127-2. PMID: 28101729.

3. Tuttle L, Meng Q, Moya J, Johns DO. Consideration of age-related changes in behavior trends in older adults in assessing risks of environmental exposures. Journal of Aging and Health. 2013; 25(2):243-273.

4. Rockwood K., Stadnyk K., MacKnight C., et al. A Brief Clinical Instrument to Classify Frailty in Elderly People. The Lancet. 1999; 353: 205-206.

5. Speechley M. and Tinetti M. Falls and Injuries in Frail and Vigorous Community Elderly Persons. Journal of the American Geriatrics Society.1991; 39: 46-52. 

6. Winograd C.H. Targeting Strategies：An Overview of Criteria and Outcomes. Journal of the American Geriatrics Society. 1991; 39: 25S-35S.

7. Lai WY. Pesticide use and health outcomes: evidence from agricultural water pollution in China. Journal of Environmental Economics and Management. 2017; 86: 93-120.

8. Ju K, Lu LY, Chen T, et al. Does long-term exposure to air pollution impair physical and mental health in the middle-aged and older adults? — A causal empirical analysis based on a longitudinal nationwide cohort in China. Science of The Total Environment. 2022; 87: 154312.

---

## [Decision Letter · Decision Letter 2]

9 Aug 2022

PONE-D-21-34489R2Agricultural Nonpoint-source Pollution and Health of the Elderly in Rural ChinaPLOS ONE

Dear Dr. WANG,

Thank you for submitting your manuscript to PLOS ONE. After careful consideration, we feel that it has merit but does not fully meet PLOS ONE’s publication criteria as it currently stands. Therefore, we invite you to submit a revised version of the manuscript that addresses the points raised during the review process.

You will find that reviewer #1 still has some minor suggestions and comments to improve the final quality of the paper.  Reviewer #1 also directly made comments in the manuscript PDF. Please carefully check those comments and address the final comments. 

We look forward to receiving your revised manuscript.

Kind regards,

Yueming Qiu

Academic Editor

PLOS ONE

Journal Requirements:

Reviewers' comments:

Reviewer's Responses to Questions

**Comments to the Author**

1. If the authors have adequately addressed your comments raised in a previous round of review and you feel that this manuscript is now acceptable for publication, you may indicate that here to bypass the “Comments to the Author” section, enter your conflict of interest statement in the “Confidential to Editor” section, and submit your "Accept" recommendation.

Reviewer #1: All comments have been addressed

Reviewer #2: All comments have been addressed

2. Is the manuscript technically sound, and do the data support the conclusions?

Reviewer #1: Yes

Reviewer #2: Yes

3. Has the statistical analysis been performed appropriately and rigorously? 

Reviewer #1: Yes

Reviewer #2: Yes

4. Have the authors made all data underlying the findings in their manuscript fully available?

Reviewer #1: Yes

Reviewer #2: Yes

5. Is the manuscript presented in an intelligible fashion and written in standard English?

Reviewer #1: Yes

Reviewer #2: Yes

6. Review Comments to the Author

Reviewer #1: The manuscript has been improved compared to last version. The "conclusion and discussion" section can be further improved. Please also go over the whole draft to make sure there are no grammar mistakes and typos. Please see more comments in the attached pdf.

Reviewer #2: The authors have addressed all my comments, and I am now convinced by the results. During the review, I have been more focused on research methods and statistical models. The authors have properly addressed all my comments and thus I have no problem with them now.

I also checked the data set availability. Although I did not download the data set myself, it seems to me that the data sources of this research are available online from Peking University. I have checked the availability on the page where they requested an application to access the data.

As for writing style, I believe the paper is now written understandably. I do believe that the other reviewer could have some comments on this part. As for me, I already feel that the manuscript is easy to read and generally understandable.

7. PLOS authors have the option to publish the peer review history of their article (what does this mean?). If published, this will include your full peer review and any attached files.

Reviewer #1: No

Reviewer #2: No

---

## [Author Response · Author response to Decision Letter 2]

19 Aug 2022

Response to Editor

Dear Dr. Qiu,

Thank you for coordinating, reviewing, and inviting revise our manuscript. We have treated yours as well as the reviewers’ comments seriously and made significant modification to the manuscript. As follows, we will summarize revisions implemented following the two reviewers’ comments before point by point responses.

For Reviewer1, we have:

1. Improved writing and readability by proving a better expression.

2. Revised the section of “conclusions and discussion” by explaining the data limitations. 

We hope you find our revised manuscript acceptable for publication in PLOS ONE.

Sincerely,

Ying Wang

Cc. Hang Xiong and Chao Chen

 

Reviewer: 1 

Thank you for the constructive comments on our manuscript. We have taken your comments seriously and substantially modified the manuscript. We believe our work has been significantly improved thanks to your comments. Please find below our response to each point you made with your original comments in italic. 

The manuscript has been improved compared to last version. The “conclusion and discussion” section can be further improved. Please also go over the whole draft to make sure there are no grammar mistakes and typos. Please see more comments in the attached pdf.

1. The “conclusion and discussion” section can be further improved. 

Response: Thank you very much for your suggestion. We have revised the section of “conclusion and discussion” by explaining the data limitations. We explored the effect of agricultural non-point source pollution on public health using fertilizer loss as a representative. We regarded animal husbandry pollution as point source pollution, but there was a small part of free range animal husbandry in rural China, which might cause non-point source pollution. Because of the limitation of data, we ignored this part of non-point source pollution. Also, loss of pesticides would also cause non-point source pollution, but there was no pesticide loss data, so we only controlled pesticides use in the empirical model. The non-point source pollution in animal husbandry and pesticides is also worth discussing if there is more wanted data.

2.Comments in the PDF documents:

Response: Thank you very much for the suggestion about typographical and grammatical errors. We have corrected the errors. We have checked the whole draft about the grammar mistakes and typos.

---

## [Editor Report · Decision Letter 3]

22 Aug 2022

Agricultural Nonpoint-source Pollution and Health of the Elderly in Rural China

PONE-D-21-34489R3

Dear Dr. WANG,

We’re pleased to inform you that your manuscript has been judged scientifically suitable for publication and will be formally accepted for publication once it meets all outstanding technical requirements.

Kind regards,

Yueming Qiu

Academic Editor

PLOS ONE
---

## [Editor Report · Acceptance letter]

5 Oct 2022

PONE-D-21-34489R3 

Agricultural Non-point source Pollution and Health of the Elderly in Rural China 

Dear Dr. Chen:

I'm pleased to inform you that your manuscript has been deemed suitable for publication in PLOS ONE. Congratulations! Your manuscript is now with our production department. 

Kind regards, 

on behalf of

Dr. Yueming Qiu 

Academic Editor

PLOS ONE